# Neurodegenerative Diseases: Can Caffeine Be a Powerful Ally to Weaken Neuroinflammation?

**DOI:** 10.3390/ijms232112958

**Published:** 2022-10-26

**Authors:** Melania Ruggiero, Rosa Calvello, Chiara Porro, Giovanni Messina, Antonia Cianciulli, Maria Antonietta Panaro

**Affiliations:** 1Department of Biosciences, Biotechnologies and Environment, University of Bari, 70125 Bari, Italy; 2Department of Clinical and Experimental Medicine, University of Foggia, 71121 Foggia, Italy

**Keywords:** caffeine, neurodegenerative diseases, neuroinflammation, immune response, neuroprotection

## Abstract

In recent years, there has been considerable research showing that coffee consumption seems to be beneficial to human health, as it contains a mixture of different bioactive compounds such as chlorogenic acids, caffeic acid, alkaloids, diterpenes and polyphenols. Neurodegenerative diseases (NDs) are debilitating, and non-curable diseases associated with impaired central, peripheral and muscle nervous systems. Several studies demonstrate that neuroinflammation mediated by glial cells—such as microglia and astrocytes—is a critical factor contributing to neurodegeneration that causes the dysfunction of brain homeostasis, resulting in a progressive loss of structure, function, and number of neuronal cells. This happens over time and leads to brain damage and physical impairment. The most known chronic NDs are represented by Alzheimer’s disease (AD), Parkinson’s disease (PD), multiple sclerosis (MS), amyotrophic lateral sclerosis (ALS) and Huntington’s disease (HD). According to epidemiological studies, regular coffee consumption is associated with a lower risk of neurodegenerative diseases. In this review, we summarize the latest research about the potential effects of caffeine in neurodegenerative disorders prevention and discuss the role of controlled caffeine delivery systems in maintaining high plasma caffeine concentrations for an extended time.

## 1. Introduction

Neuroinflammation represents a defense mechanism aimed to protect the brain by removing or disrupting noxious agents and microorganisms [1]. Although this host mechanism seems to determine beneficial effects—by removing cellular debris and promoting tissue repair as well as preserving the brain integrity—prolonged and sustained inflammation may result in some circumstances that are detrimental, causing damage to nervous tissue, thereby leading to neuron death and developing neurodegenerative diseases. [2,3]. Neuroinflammation is the inflammatory response described in the brain and involves some glial cells, named microglia and astrocytes, which actively participate to innate immune response in the central nervous system (CNS). However, if microglia and astrocytes remain activated for too long, they become responsible for a persistent inflammatory response, which can cause neurodegenerative disorders [2].

Within the last two decades, significant advances regarding the role that microglial cells play in the development of CNS diseases have been reached, these advances demonstrate how microglia activation is a trademark of a wide array of neurodegenerative diseases, such as Alzheimer ‘s disease (AD), Parkinson’s disease (PD), multiple sclerosis (MS), as well as other brain diseases including amyotrophic lateral sclerosis (ASL) and Huntington’s disease (HD) due to the dysregulation of the defenee function and neuroinflammation [4,5].

Therefore, the deletion of negative effects associated with microglial activation have emerged as a possible therapeutic strategy to counteract the neuroinflammation-associated neurodegeneration [6,7].

Unfortunately, until now, no drug has been described capable to block or slow the progression of neurodegenerative pathologies, thus, actually a high number of investigations focalize the attention to search for natural bioactive compounds that could have beneficial effects on brain disorders, without affecting healthy cells.

Caffeine (1,3,7-trimethylxanthine) (Figure 1) represents the most widely consumed psychostimulant, being one of the major components of coffee, tea and energy drinks. It was estimated that around 166.63 million 60 kg bags of coffee were consumed worldwide in 2020/2021. Moreover, global coffee consumption is estimated to increase [8]. The most important physiologically effective bioactive compounds present in coffee include several antioxidants such as chlorogenic acid, lignan, melanoids, cafestrol, trigonelline, kahweol and caffeine [9].

Caffeine intake from all sources is estimated to be between 70 and 350 mg/person/day. Moreover, a single cup of coffee contains between 0.4 and 2.5 mg/kg of caffeine, which is absorbed through the small intestine in about 45 min, resulting in a peak blood concentration of 0.25 to 2 mg/L, or around 1 to 10 μM [10].

Caffeine has a half-life of 3 to 7 h in healthy adults, and it is primarily metabolized in the liver by the cytochrome P450 oxidase enzyme system (CYP1A2 isozyme) into three dimethylxanthines, including paraxanthine, theobromine and theophylline, all of which are pharmacologically active in the human body, and then secreted in the urine [11].

Moreover, caffeine has been described in all body fluids including saliva, bile, plasma, cerebrospinal fluid, semen, umbilical cord blood and breast milk [9]. It was reported—in an in vivo rat model—that caffeine is present in all tissues after administration for 10 days and accumulated for 25 days. In this context, the caffeine level resulted in significantly higher levels and was widely distributed in brain, liver and kidneys [12]. Caffeine is water- and lipid-soluble; therefore, it can readily cross the blood brain barrier and, once in the brain, it may act on different molecular targets to determine multiple pharmacological effects, such as to antagonize with adenosine receptors, inhibit phosphodiesterase and block calcium release [13]. In this context, it has been demonstrated that caffeine does not alter the blood–brain barrier [14], thus, leading to no alteration in brain parenchyma physiology and homeostasis. However, on the other hand, energy drinks are able to induce blood–brain barrier dysfunction, as recently demonstrated [15], and this side effect could explain the previously observed brain oxidative stress in rats under an energy drink condition [16]. Now, it is increasingly evident that non-toxic doses of caffeine mostly act on biological tissues by the antagonism of adenosine receptors, as first stated by Fredholm [17]. Furthermore, Lopes et al. highlighted that the antagonism of adenosine receptors is the main mechanism operated by non-toxic doses of caffeine to modulate activity in brain circuits. The same study also showed that the effect of moderate concentrations of caffeine on the control of synaptic transmission and plasticity in the mouse hippocampus is selectively mediated by antagonizing adenosine receptors, where A1R and A2AR control basal synaptic transmission and synaptic plasticity, respectively [18].

Several studies have highlighted the link between coffee consumption and improved health, as described by an increasing number of systematic reviews and meta-analyses, which report that coffee consumption is associated with a lower risk of several chronic pathologies related to inflammation processes, including neurodegenerative conditions such as AD [19,20,21,22]. It was reported, in some animal models, that whereas acute administration of caffeine exacerbates neuronal damage in experimental ischemia, low or chronic doses are able to protect CNS from hypoxia and ischemia, thus, hypothesizing that the regular consumption of caffeine, at a low dose and for prolonged times, could be helpful in preventing neurodegenerative diseases [23,24].

This manuscript reviews the recent insights on the role of caffeine consumption as a nutrient-based preventive strategy for food supplementation to counteract oxidative stress and neuroinflammation in common neurodegenerative diseases.

## 2. Neuroinflammation, Neurodegeneration and Caffeine

Neuroinflammation is a complicated process involving the integration of immediate local inflammatory responses by all CNS cells, including neuronal cells, macroglia and microglia. Based on this, immune cells such as macrophages, T cells and B cells are recruited throughout the body [25].

Some factors—such as genetic background, environmental factors, a possible initiating insult and age—can work together to activate microglial cells that are so entangled in the complex neuroinflammatory pathway [26,27]. All of these variable factors can cause inflammatory reactions, which begin by intracellular signaling cascades shortly after cellular injury and tissue damage. Lipopolysaccharide (LPS), an endotoxin found in the outer membrane of gram-negative bacteria, for example, causes systemic inflammatory response syndrome via toll-like receptor (TLR) signaling activation [28].

When LPS binds TLR-4 on the microglia surface, in fact, several signal transduction pathways are activated—such as PI3K/AKT, MAPK and mTOR—which, in the end, are responsible for NF-kB activation. In this respect, NF-kB activation plays a pivotal role in the production of chemokines, pro-inflammatory cytokines, inducible enzymes, such as inducible nitric oxide synthase (iNOS), and ciclooxigenase (COX)-2, which all together can result in neuroinflammation [29].

Neuroinflammation is usually beneficial for the proper control of external stressors. Nevertheless, prolonged (or chronic) immune response due to aging or to immunosenescence can lead to a dysregulation of immune signals, thereby leading to neurodegenerative pathogenesis [30,31]. Many neurodegenerative disorders are characterized by neuroinflammation. In this context, neurodegenerative disease is an umbrella term that refers to a number of conditions that predominantly impact neurons in the human brain. Neurodegenerative diseases are chronic disorders mostly caused by a generalized dysfunction of brain homeostasis, which results in a progressive loss of function, structure and number of neurons, which causes brain damage or physical dysfunction over time [32]. These diseases are associated with impaired central, peripheral and muscle nervous systems [33]. Neurodegenerative diseases are debilitating and non-curable diseases that cause the progressive degeneration as well as the death of nerve cells from which movement disorders, called ataxias, or mental functioning, called dementias, ensue. Neurodegenerative diseases are divided into acute and chronic. Acute conditions can cause permanent nerve damage over a short period of time, while chronic ones, including AD, PD, MS and ALS, hamper the quality of life of the elderly. The number of people over the age of 65, in Western countries, has steadily increased, consequently increasing the risk of age-related neurodegenerative diseases. The most common pathology is AD, which constitutes 50–70% of total neurodegenerative diseases and nowadays more than 26 million people are affected from this worldwide disease, a number that is expected to quadruple by 2050 [34]. PD is the second most common neurodegenerative disease and is characterized by a high morbidity among middle-aged and elderly people. There are no effective treatments for aging-related neurodegenerative diseases, which tend to irreversibly progress and be associated with dementia, with huge personal and socioeconomic costs [35].

In this respect, it is well known that neurons are affected in aging, particularly in some brain areas such as the neocortex and hippocampus, as evidenced by the significant decline in motor, sensory and cognitive functions [36].

This is due in part to the fact that, despite its tiny size (around 2% of body weight), the brain consumes approximately 20% of cellular energy and generates a large quantity of metabolic waste. If these wastes are not adequately removed, they accumulate and cause significant neuronal damage, which could eventually dissolve its integrity, causing neuronal aging and a range of neurodegenerative diseases [37]. Another aspect of neuronal vulnerability that is linked to aging, is the fact that degenerated neurons are irreplaceable, since the replacement of neurons through neurogenesis or the bypassing of damaged neurons through neural rewiring is a very limited event. In the context of neurodegenerative diseases, existing therapies can only limit and slow down disease progression. It is therefore imperative to look for ways to protect neurons before they die, by preventing or overcoming age-related neuronal damage, to reduce the chance of developing disease [38]. The primary problems of scientific research are the prevention of these pathologies and the discovery of new nutraceuticals and medications to counteract them. Plant-derived compounds, especially caffeine, are known to have a number of protective effects related to neurodegeneration, including anti-inflammatory and antioxidant properties [39].

## 3. Caffeine and Glial Cells

It is well known that the glial cells such as microglia and astrocytes, in the normal physiological CNS state, sustain brain homeostasis, neuronal development and provide defense of the brain from injury, infection and damage. Compared to other glial cells and neurons that develop from the neuroectoderm, microglia originate from the myeloid hematopoietic lineage and are distributed in the brain with wide regional differences [40,41]. Microglia are the primary resident innate immune cells of the brain parenchyma and represent the most effective component in neuroinflammatory response regulation. It is evident that microglia have a dual-action: pro-inflammatory and anti-inflammatory actions. A mild activation of microglial cells in response to injury, stress or inflammatory stimuli is known to induce a protective cellular response that involves tissue repair and a return to homeostasis. A chronic or prolonged reactivity of these cells, instead, may activate inflammatory responses that contribute to the death and dysfunction of neurons, which appears to occur in neurodegenerative diseases, via the release of various pro-inflammatory and neurotoxic factors, such as tumor necrosis factor alpha (TNF)-α, interleukin (IL)-1β, chemokines, proteases and various free radicals [42,43].

In recent years, a large body of evidence reported that, in addition to microglia, astrocytes are also involved in neuroinflammatory responses and that they can play essential roles in health and disease [44]. Astrocytes are abundant cells in the brain and are highly heterogeneous in form and function. Moreover, they show a remarkable adaptive plasticity that allows them to maintain the integrity of the neural microenvironment. Therefore, astrocytes are involved in a wide range of homeostatic functions, which include others such as the regulation of brain development, structural and trophic support for neurons, clearance of neurotransmitters and regulation of synaptic activity and, lastly, the formation and maintenance of the blood–brain barrier (BBB) [45]. Similar to microglia, astrocytes respond to neuronal damage with two different phenotypes: one is associated with neuroinflammation (A1), while the other one is associated with neuroprotection (A2). In the context of inflammatory brain disorders, the cooperation of both glial cells represent—through their increasing reactive response—the main defensive system of the CNS. Thus, an appropriate crosstalk between astrocyte and microglia in pathological conditions is necessary for neuronal homeostasis after an insult. During pathological conditions, in fact, a glial cell’s irregular activation and the development of neurotoxic factors may result in a chronic neuroinflammatory state that is able to induce neuronal death [46].

Several lines of evidence report that the modulation of the excessive activation of glial cells by caffeine and its components can limit the neuroinflammation associated with neurodegenerative disorders [47]. In male Swiss albino mice co-treated intraperitoneally with paraquat and maneb, both pesticides were able to induce neurodegeneration and motor deficits characteristic of PD; the caffeine treatment was able to reduce the levels of nitric oxide (NO) and the number of activated microglial cells by protecting dopaminergic neurons. The same study suggests that caffeine-mediated neuroprotection is regulated by NF-kB, tyrosine kinase and p38 mitogen-activated protein kinase pathways [48].

Figure 2 summarizes the protective effects of caffeine consumption in nervous tissue. Several studies have reported the protective effects of caffeine against LPS-induced neurodegeneration.

In an in vitro investigation, conducted on the murine BV-2 microglial cell line, 2 mM caffeine inhibited the production of proinflammatory mediators such as NO, prostaglandin-2 (PGE_2_) and TNF-α, as well as their regulatory genes, which were triggered by LPS. This suggests that caffeine is able to induce anti-inflammatory effects through an Akt-dependent NF-kB and ERK (extracellular signal-regulated kinase) pathway [49]. Another study evaluated the effects of caffeine against LPS-induced inflammation and neurodegeneration in adult mouse brains. The authors demonstrated that a daily injection of caffeine, for four weeks, markedly reduced the expression of TLR-4, calcium-binding adapter molecule 1 (Iba-1) and glial fibrillary acidic protein (GFAP), which are, respectively, the cellular markers of activated microglia and astroglia, as well as the expression of phospho-NF-kB and phospho-c-Jun n-terminal kinase (p-JNK) upregulated in the LPS treated mice. Furthermore, the findings of this work suggested also that caffeine may be able to reduce elevated oxidative stress by the regulation of the levels of nuclear factor erythroid-2-related factor 2 (Nrf2) and of the enzyme hemeoxygenase 1 (HO-1) in caffeine and LPS co-treated mice [50]. Similar results were observed when D-galactose-treated rats were treated with chronic caffeine, showing that caffeine seems significantly able to suppress p-JNK -induced neuroinflammation and neurodegeneration [51]. Khan et al. have reported that caffeine may ameliorate cadmium-induced neurotoxic effects, which is a non-biodegradable heavy metal, as it results both in adult mouse brains and in HT-22 and BV-2 cell experiments. These studies showed that caffeine mitigated the cadmium-mediated activation of glial cells by suppressing GFAP and Iba-1 inflammatory marker expression in the adult mouse cortex and hippocampus. The authors suggest that the neuroprotective effects of caffeine against cadmium-induced neurotoxicity were mediated by Nrf-2 and NF-kB signaling regulation in the HT-22 and BV-2 cell lines, respectively [52]. The cytoprotective effect of caffeine, moreover, was observed also in an in vitro PD model against 6-hydroxydopamine (6-OHDA)-induced cytotoxicity in rat mesencephalic cell cultures. Caffeine reduced, in a significant manner, the number of activated microglia and reactive astrocytes after cells were exposed to caffeine and neurotoxin simultaneously [53]. Similar results were obtained in PD-like pathology in α-synuclein (Syn)-induced mice. In this respect, chronic caffeine treatment significantly reduced α-Syn-induced microglial activation, as well as reactive astrogliosis in striatum [54]. It is reported in some animal models that the role of caffeine may be neuroprotective or neurotoxic depending not only on the dose, but also on the time of administration. The chronic administration of caffeine at low doses prevented the expression of CD11b and GFAP attenuating brain damage induced by 3,4-methylenedioxymethamphetamine (MDMA) in male mice striatum. The researchers suggest that chronic consumption of caffeine for 21 days may develop tolerance to caffeine and induce an increase of body temperature. Therefore, the lack of a rise in temperature after NMDA in mice chronically treated with caffeine may reduce glial neuroinflammation [55]. The protective action of caffeine was also observed in a neuroinflammation model experimentally induced by chronic infusion of LPS over a period of two up to four weeks in the brain of young F-344 male rats. The study highlighted that a daily intraperitoneal injection of caffeine reduced the LPS-induced number of activated microglia in the hippocampus, in dose- and time-dependent manners [56]. In addition, Sonsalla and coworkers demonstrated that chronic caffeine administration attenuated the microglial activation in the substantia nigra *pars compacta* (SNpc) of rats infused into the left cerebral ventricle with 1-methyl-4-phenylpyridinium (MPP+). The authors suggest that the neuroprotective actions of caffeine in the SN, as an adenosine receptor antagonist, could be due to caffeine competitive inhibition with adenosine [57]. In a cortical-impact model of TBI in mice, acute and chronic caffeine treatment attenuated inflammatory cell infiltration with a significant reduction in CD45-positive cells, which are a marker of pro-inflammatory cells such as microglia, macrophages and neutrophils, in the damaged areas of the brain of mice chronically treated with caffeine. In addition, chronic caffeine treatment also suppressed inflammatory cytokine production such as TNF-α and IL-1 [58]. This is in contrast with Khairnar et al., who reported that caffeine acutely administered with MDMA at doses similar to those that may be taken for recreational use, assuming energy drink-like levels was able to induce neuroinflammation in the brains of adult male C57BL/6J mice, characterized by microgliosis and astrogliosis in the caudate–putamen (CPu). On the contrary, in SNpc, caffeine did not affect MDMA-induced glial activation. Thus, the combination of NDMA plus caffeine can facilitate NMDA-induced inflammatory processes in a mouse brain [59]. In a subsequent study, the same group showed that in CPu sections of adolescent mice, the combined treatment with MDMA plus caffeine enhanced both NDMA-induced astroglial activity (GFAP reactivity) without significantly altering the microglial activation (CD11b reactivity) and the levels of TNF-α and IL-β. This study suggests that using MDMA with caffeine throughout adolescence may worsen the neurotoxicity and neuroinflammation caused by MDMA [60]. The benefits of caffeine were also detected in an animal model of glaucoma. The caffeine administration prevented microglia activation and reduced levels of inflammatory mediators both in eyes induced by ocular hypertension (OHT) and in contralateral retina, without OHT, affording protection to retinal ganglion cells [61]. In a subsequent study, the same group of researchers demonstrated the effect of caffeine, as an adenosine receptor antagonist, in a rat model of retinal transient ischemia. This research revealed that caffeine consumption had a variable effect on microglia reactivity depending on the reperfusion duration. Caffeine was shown to suppress microglia activation and reduce both IL-1 and TNF-α levels at later time points upon injury (7 days of reperfusion). The authors propose that excessive microglia activation at an earlier time point is important in the control of inflammation and that persistent caffeine treatment is protective for the retina [62].

A recent investigation examined the mechanism underlying the neuroprotective role of caffeine in an animal model of sepsis-associated encephalopathy (SAE), demonstrating that caffeine, in the rat cerebral cortex, is able to reduce neuronal apoptosis and astrocytic activation, as well as decrease ROS production and mitochondrial dysfunction, thereby reducing SAE by inhibiting the UCP2-mediated NLRP3 pathway in astrocytes. Furthermore, in the same study, the authors also showed that caffeine was able to reduce the levels of NLRP3, IL-1 and IL-18 in in vitro primary astrocytes isolated from rat cerebral cortex treated with LPS and IFN-γ, thereby confirming the possible protective role of this molecule during brain inflammation via the modulation of glial cells activation [63]. Another recent investigation demonstrated that caffeine had anti-inflammatory properties in newborn rats with hypoxia–ischemia-induced cerebral white-matter injury via NLRP3 inflammasome suppression via the adenosine A2a receptor (A2AR). The authors demonstrated that caffeine treatment abolished NLRP3 inflammasome, decreased microglial Iba-1 activation and the pro-inflammatory M1 phenotype by downregulating CD86 and iNOS protein expression and inhibiting associated TNF-α and IL-1β secretion. Caffeine, at the same time, was also shown to upregulate the anti-inflammatory M2 phenotype, increasing CD206 and Arg-1 expression and suppressing the release of anti-inflammatory cytokines such as IL-10 and transforming growth factor-β (TGF-β). Interestingly, the researchers additionally reported that inhibition of the A2AR reversed these caffeine-mediated activities [64].

The protective role of caffeine was also evaluated against disruption of the BBB that occurs in a variety of neurological disorders [65]. In a rabbit model of sporadic AD, Chen and collaborators reported that a chronic caffeine administration to rabbits on a cholesterol-enriched diet avoided BBB dysfunction, decreasing the expression of the endothelial cell tight junction proteins occludin and zonula occluden, thereby diminishing the activation of astrocytes and activated microglia density in rabbit brains [66]. The same authors obtained similar results in an MPTP neurotoxin model of PD [67].

## 4. Neuroprotective Effect of Caffeine in Neuroinflammatory and Neurodegenerative Diseases

A number of studies suggest a protective effect of caffeine on multiple neurodegenerative and neuroinflammatory conditions. Caffeine, in fact, has powerful antioxidant, anti-inflammatory, and anti-apoptotic actions against several types of neurodegenerative diseases, including AD, PD and other neurodegenerative disorders (Figure 3).

### 4.1. Neuroprotective Effect of Caffeine in AD

AD is the most common form of dementia. This disorder, typical of elderly, is a neurodegenerative syndrome characterized by a slow and permanent loss of cognitive function. Memory loss, language issues, personality changes, lack of initiative, confusion, disorientation and the loss of logic and judgment are the most common symptoms of AD. Experts think that hereditary and environmental variables, as well as a particular style and, in some cases, familiarity with the condition, all contribute to the development of AD. The evidence on the pathophysiology of this form of dementia is clear: The cerebral extracellular deposition of diffuse and neuritic senile plaques is made by Aβ peptides, as well as the intracellular aggregation of flame-shaped neurofibrillary tangles (NFTs) that are made up of hyperphosphorylated aggregates of the microtubule-associated tau protein selectively mediate large-scale neuronal loss.

The tau-hyperphosphorylated protein and beta-amyloid (specifically the variant Aβ42) are proteins produced by the brain, the latter of which the brain is unable to eliminate. Both of these proteins then accumulate and start damaging neurons (the cells of our brain) many years before memory disturbances appear [68,69]. Cell death begins in a region of the brain called the hippocampus. The hippocampus, located in the temporal lobe, is primarily involved in learning and memory processes. Subsequently, cell death extends to the involvement of the entire brain and leads to additional cognitive and functional difficulties that are observed in people with AD [70].

The mechanism behind the production of amyloid plaques and neurofibrillary tangles remains unknown. What is known is that they involve the damage and death of brain cells, resulting in memory difficulties and behavioral changes. Further, hypotheses include the presence of beta-amyloid oligomers which, like the amyloid plaques, are also potentially neurotoxic. Furthermore, the abnormal release of neurotransmitters, such as glutamate, also contributes to neuronal death and inflammatory processes within the brain. The neuro-inflammatory process is similarly involved in the complex cascade of processes that cause AD and subsequent symptoms. This process is therefore implicated both in the pathogenesis of AD and in its progression [68].

Currently, there have not been described efficient pharmacological approaches capable of reversing cognitive impairment and typical decline associated with AD.

Previous studies report that caffeine could prevent the Aβ aggregation through hydrophobic contacts with monomers or small aggregates [71].

Successive investigations demonstrated detectable effects of caffeine on Aβ fibrillization only when it was used at higher doses (10-fold molar excess caffeine). These surprising discrepancies probably reflect the different experimental conditions, being the prior existing study, which employed truncated Aβ_16-22_, instead of Aβ_1-40_ used in the other investigations, thus, highlighting the complexity of both assay conditions and aggregation processes. For example, changes in the pH solution, buffer composition, as well as differences of Aβ or tau concentrations, may lead to variations in the experimental outcomes [72]. The reported results have been conducted in in vitro models; therefore, it is possible that the same effect may not be reproduced in in vivo system. In this regard, Costa et al. reported that caffeine treatment in aged mice results in a significantly higher recognition and memory capacity than in age-matched control mice; this action may be due to a neuroprotective effect linked to the upregulation of the brain-derived neurotrophic factor (BDNF) or that of the tyrosine kinase receptor (TrkB) at hippocampal level [73]. Confirming these observations, Arendash et al. showed that aged transgenic AD mice fed an equivalent of 5 cups of coffee per day, in humans, had lower levels of β-amyloid in the hippocampus because of the suppression of both β-secretase (BACE1) and presenilin 1 (PS1) γ-secretase. Furthermore, these authors have shown, regarding the mechanism of action, that the suppression of caffeine by BACE1 involves the cRaf-1/NF-kB pathway [74]. Finally, aged APPsw mice showed improvement in memory capacity and a reversion of AD pathology, thus, suggesting a possible therapeutic role for caffeine in determined AD cases [75].

Some studies emphasized that the intake of caffeine prevents memory impairment in animal models of AD. Dall’Igna et al., in fact, demonstrated that combined acute (30 mg/kg) and prolonged (1 mg/mL) treatments of caffeine prevented cognitive dysfunction induced by 25–35 fragments of β-amyloid in mice, an effect mimicked by selective A2A receptor antagonists. The same outcome was also obtained through subchronic (4 days) treatment with daily injections of caffeine (30 mg/kg) [76]. Other authors observed, in mice, that caffeine consumption at higher non-toxic doses improved the performance of β-amyloid-induced spatial memory resulting from the blockade of the A2A receptor. However, the same neuroprotective effect was also observed combining caffeine-prolonged treatment with the acute administration of caffeine This treatment, in fact, seems to be able to improve the behavioral effects resulting from the blocking of the A2A receptor, while causing tolerance to the effects of the A1 receptor blockade [77]. In another study, Canas et al. showed that the pharmacological or genetic blockade of the A2A receptor prevents the synaptotoxicity induced by Aβ1-42 and the consequent memory deficit through a p38—MAPK (mitogen-activated protein kinase)-dependent pathway. This report provides a molecular basis for the benefits of caffeine consumption in AD [78]. Furthermore, in an experimental model of sporadic AD, it was shown that caffeine consumption prevented streptozotocin (STZ) -induced behavioral modifications and neurodegeneration in the hippocampus, as well as upregulation of the A2A receptor [79].

An analytical study of the risk factors for AD, conducted on a group of 1023 people aged 65 and over, found that coffee consumption is associated with a 31% reduction risk of disease incidence [80]. This result has also been described by Eskelinen and collaborators who evaluated the association between coffee consumption in middle age and the risk of AD/dementia in old age over a 21-year follow-up. Consequently, moderate coffee users (3–5 cups/24 h) were shown to have a lower risk of AD/dementia than low coffee users (0–2 cups/24 h), thus, suggesting that regular coffee/caffeine can be beneficial for both AD and dementia [81].

### 4.2. Neuroprotective Effect of Caffeine in PD

About 1% of people over the age of 60 worldwide are severely affected by PD, which is the second most common neurodegenerative disease and is caused by dopaminergic neuronal loss of the SNpc. PD is a neurological disorder in which the first symptoms are not progressively evident, so, as a consequence, patients often lack adequate timely treatment [82,83]. PD produces both motor and non-motor impairment. Motor strength appears after the loss of 50–70% of nervous system cells and includes rigidity, bradykinesia, resting tremors and inadequate postural reflexes. Non-motor symptoms include abnormalities of mood, cognitive function, sleep, autonomy, dementia and changes in smell and memory. This condition is recognized to be etiologically caused by both environmental (90–95%) and genetic (5–10%) causes [84].

Many causative factors associated with PD have been identified so far, such as dopamine metabolism, impaired mitophagy, electron transport chain dysfunction, the induction of aberrant neuroinflammation, oxidative stress, activation of microglia, the formation of Lewy bodies and aging. Lewy bodies are mainly given by the deposition of α-syn encoded by the SNCA gene that plays a pivotal role in the pathogenesis of PD. The abnormal accumulation of soluble monomers of α-syn leads to oligomers formation and fibrils as a central event in the early stages of PD [85].

Many metabolic activities required by the organism result in oxidative stress [86]. On the other side, it has the potential to be harmful and detrimental to the body. Oxidative alterations, such as a reduction in the antioxidant defense system or the activation of glial cells, which are a source of oxidative stress, are crucial in the pathophysiology of PD because the generation of reactive oxygen species during the progression of PD damages the substantia nigra by lipid peroxidation, protein oxidation and DNA oxidation. Changes in the antioxidant defense system appear to be the cause of this condition. As a result, it is recognized that oxidative stress and inflammation are two essential variables that contribute to nervous system damage [87,88].

Currently, therapeutic treatments involve the administration of drugs that alleviate symptoms. As a result, new treatments are required, not just to arrest its development, but also to prevent it. For these reasons, natural substances with neuroprotective effects, including coffee, have gained attention. Caffeine has neuroprotective qualities, which may be connected to its antioxidant characteristics [89]. Caffeine can also reduce lipid peroxidation by lowering the generation of reactive oxygen species such as hydroxyl radicals and hydrogen. It can also serve as an antioxidant by increasing glutathione S-transferase activity. Specifically, the high activity of monoamine oxidase-B (MAO-B), present in PD catalyzes the oxidation of dopamine, thereby generating H_2_O_2_. In this way, increased oxidative stress is responsible for the loss of dopaminergic neurons. Caffeine, on the other hand, has recently been demonstrated to have neuroprotective benefits by blocking MAO-B, which may favor an increase in dopamine levels and so ameliorate motor symptoms [90]. Moreover, it seems that caffeine can also protect dopaminergic neurons by the activation of some antioxidant signaling molecules, such as the erythroid-related nuclear factor 2/Keap1 and the coactivator 1α of the receptor, activating the proliferation of gamma peroxisomes, promoting the activation of transcription factors, which are involved in the biogenesis of mitochondria, as well as in antioxidant and anti-inflammatory pathways [91].

Furthermore, the caffeine neuroprotective effect is also supported by animal studies, showing that this molecule confers neuroprotection against dopaminergic neurodegeneration both in neurotoxin PD models, where mitochondrial toxins (MPTP, 6-OHDA and rotenone) are used, and in an α-syn transmission mouse model through the intracerebral injection of α-Syn fibers [13,54,92]. It is worth noting that in a chronic MPTP infusion model of PD, caffeine can give protection against dopamine neurodegeneration even after the neurodegenerative process has started (i.e., 14 days after MPTP infusion) [57]. In addition, a recent study revealed that coffee might protect against α-syn-induced pathological alterations in an A53T animal model of PD and that this effect could be related with increased autophagy activity [54]. Importantly, the pharmacological blockade or genetic deletion of the adenosine A2AR—the main pharmacological target of caffeine in the brain—seems to protect against dopaminergic neurodegeneration in PD animal models [93]. This observation suggests that caffeine protective effects are probably due to its action on this receptor. Moreover, A2AR modulated α-syn aggregation and toxicity in SHSY5Y cells, as well as A2AR blockade, and was able to rescue synaptic and cognitive deficits in α-syn transgenic mouse model of PD, thereby showing that caffeine consumption can lower the risk of developing PD and supporting the clinical potential of caffeine and A2AR antagonists as a disease-modifying drug target for this condition [94,95].

In recent decades, several studies have been conducted to evaluate the effects of caffeine as a nutraceutical compound in PD. In a double-blind, controlled phase 2/3 complete study (NCT00459420), in fact, the effects of caffeine in idiopathic PD patients were evaluated. The main aim of this study was to evaluate caffeine efficacies for excessive daily sleepiness in PD. All the participants continued to take their PD medication and were asked to drink caffeinated beverages. Caffeine was well tolerated and had no negative side effects. Despite an improvement in motor manifestations, it was seen that caffeine seems to have no effect on excessive daytime somnolence in PD patients [96]. The potential motor benefits, due to caffeine consumption, were better explored in a larger long-term trial (NCT01738178). In this trial, patients affected by idiopathic PD received 200 mg caffeine. In comparison to the control group, caffeine administration resulted in no adverse events. According to the findings of this research, caffeine did not improve motor function in PD patients, so future study is necessary to determine the epidemiological linkages between caffeine use and a reduced incidence of PD. Several prospective studies were carried out in order to reveal the favorable benefits of caffeine in PD [97]. A prospective investigation was, in fact, performed to assess the potential link between caffeine intake and the risk of PD in men and women. This study concluded that low levels of coffee drinking lowered the risk of PD in men more than in women [98].

Another study conducted on a large cohort of men and women revealed that caffeine consumption reduced the risk of developing PD in both sexes [99]. In this regard, recent reports showed that caffeine consumption was lower in PD patients compared to healthy subjects, and a high caffeine level was associated with a lower risk of idiopathic PD in a sex-independent way. As a result, habitual caffeine consumption may benefit humans by lowering the risk of PD [100].

In a large cohort of 8004 Japanese American males who were tracked for 30 years, the relationship between regular caffeine use and a decreased risk of PD was also explored. During this investigation, Ross et al. discovered a decreased risk of PD proportionate to coffee usage. Furthermore, non-coffee users had a fivefold increased risk of PD compared to those who drank 28 ounces or more per day [101].

Therefore, numerous supports emerge from the literature for the neuroprotective value of caffeine intake in patients with PD. The possible mechanism of action responsible for the protective effect of caffeine remains poorly studied. This mechanism should certainly be investigated in order to better understand the therapeutic potential of caffeine-based therapies in PD.

### 4.3. Neuroprotective Effect of Caffeine in MS

MS is degenerative disorder of the CNS characterized by chronic inflammation. Presumed to be autoimmune and characterized by demyelinating processes, MS occurs in the white and grey matter of the CNS and affects people of all ethnicities, ages and sexes. It is estimated that it affects approximately 400 000 people in the United States and 2.5 million people worldwide. The reason behind the development of MS has not been discovered yet, but it has been largely reported that not only genetic causes, but also environmental factors, may contribute to its development. [102,103]. MS causes inflammation and demyelination, thus, resulting in lesions in the white and grey matter. The symptoms of this pathology are caused by a decrease in conductance and/or blockage, together with axonal injury and neuron death. Moreover, symptoms may include eye problems, numbness, brain stem symptoms, bladder dysfunction, ataxia, paresis and a slowly increasing cognitive disability, depending on the lesions position [104,105].

Given the availability of 15 officially authorized disease-modifying therapies, the majority of MS patients keep suffering disability accrual and chronic symptoms, underlining the need for additional therapies. Unfortunately, MS is not a curable disease, but it is known that its course may be very positively influenced by medication [106].

Research has shown that the anti-inflammatory effect of caffeine can be involved in reducing the likelihood of developing MS, but the mechanism responsible for this is still unclear. In reference to MS, the neuroprotective properties of caffeine can suppress inflammation and assist with symptoms such as constipation and cognitive fog [107].

The protective effect of caffeine in MS has mainly been evaluated in clinical studies. In particular, in recent years, a study revealed a significant association between high consumption of coffee and a decreased risk for MS; in fact, a cross-sectional survey showed that coffee consumption, at least in the relapsing form of MS, has positive effects on the progression and disease course, when comparing daily coffee drinkers to the non-daily coffee drinkers [108]. An experimental study was conducted, in addition, to determine whether chronic caffeine treatment has any neuroprotective effects on the course of disease in an animal model of MS. In this regard, an EAE rat model, distinguished by widespread tissue inflammation and a chronic disease course, was used, and it was demonstrated that the incidence of EAE was reduced in rats treated with caffeine. The same study also showed that the disease was mitigated at histological, neurochemical and behavioral levels when compared to rats given only water [109].

Another study in EAE supports that caffeine intake is associated with a lower risk of developing MS. According to this research, the caffeine protective effect appears to be due to an improvement in the integrity of the BBB, possibly via the caffeine-induced activation of adenosine 1A receptors (A1). Recently, two important case-control studies on coffee intake and MS indicated that coffee consumption was related with a lower possibility of developing MS in a dose-dependent way. Finally, a more recent study showed that coffee consumption may be a therapeutic approach for selected individuals with MS-related fatigue in the absence of a successful fatigue treatment [110,111,112].

Despite all these observations, few works about caffeine supplementation as a therapeutic modality for MS are present in the literature. In this respect, very recently, it was reported that caffeine, in an experimental autoimmune encephalomyelitis (EAE) animal model, is able to diminish NLRP3 inflammasome activation, thereby determining a neuroprotective effect by inducing autophagy and reducing both the infiltration of inflammatory cells and demyelination. In the same work, it was also shown that, in an in vitro model of neuroinflammation, caffeine was able to inhibit the activation of the NLRP3 inflammasome by promoting autophagy and by suppressing the mechanistic target of mTOR pathway [113].

Therefore, although the protective effect of caffeine is reported regarding MS, the cellular mechanism involved in the protective responses to the action of this molecule has not yet been investigated.

### 4.4. Neuroprotective Effect of Caffeine in ALS

ALS is an incurable and rapidly progressing neurodegenerative disease, which is characterized by a progressive degeneration of motor neurons in the spinal cord and motor cortex, resulting in skeletal muscle atrophy and leading to death by respiratory failure within 3–5 years of initial symptoms [114]. Although most cases of ALS are sporadic, the disease can occasionally also be caused by a single gene mutation, such as the Cu^2+^/Zn^2+^superoxide dismutase (SOD)-1 mutation [115].

Regarding the cellular level, instead, an excessive stimulation of glutamate receptors seems to lead to a large influx of calcium ions into the postsynaptic neuron, which results in oxidative stress, oxidative damage, inflammation and apoptosis [116].

Several studies suggested that caffeine seems to be able to play an important modulatory role in ALS via a variety of mechanisms. In this respect, the excitatory neurotransmitter glutamate has been suggested to play a role in ALS. It is known that chronic neuroinflammation is linked with an increase in extracellular glutamate levels. Drugs that limit glutamate effects on neuronal receptors have been shown to reduce, indirectly, the neuroinflammatory response of microglia cells. Interestingly, in fact, caffeine attenuated the number of activated microglia within the hippocampus of animals with LPS-induced and age-related inflammation [56]. Moreover, it was also reported that maternal caffeine intake during gestation is able to determine the downregulation of A1 and metabotropic glutamate receptors in the brain of both rat mothers and fetuses [117].

In addition, as previously reported for PD, it was also found that A2AR inhibition is beneficial in the SOD1G93A mouse model of ALS. A2AR is normally abundantly expressed in spinal cord cells, including motor neurons. A2AR levels in the spinal cord of G93A mice are higher than in wild-type mice, but daily A2AR antagonist treatment appears to be highly effective in increasing motor neuron survival, slowing the progressive loss of forelimb grip strength, delaying disease onset and, finally, extending overall survival [118].

Despite the observations reported above, a large longitudinal study revealed no connection between caffeine consumption and ALS risk, and, at the same time, the results of this study do not support the hypothesis that caffeine consumption is associated with a decreased risk of ALS [119].

### 4.5. Neuroprotective Effect of Caffeine in HD

The inherited neurodegenerative disorder name HD is determined by expanded CAG repeats and it is characterized by cognitive and psychiatric disturbances, as well as by motor symptoms. HD is, in fact, a hyperkinetic disorder in which the main symptoms are represented by chorea (jerky, involuntary movements), tremor, dystonia, which manifests itself with abnormal muscle tone resulting in muscle spasm and abnormal posture, and prominent neuropsychiatric and cognitive changes [120]. During HD, the signs of neurodegeneration appear in several cerebral regions of the brain, but the primary neuropathological hallmark is represented by the atrophy of the striatum, with a particular involvement of striatopallidal neurons expressing dopamine receptors [121]. Chronic quinolinic acid (QA) lesions in rats are very similar to the neurodegeneration seen in HD [122]. QA intrastriatal administration can cause an overstimulation of NMDA receptor which, in turn, determines an influx of Ca^2+^ and that, eventually, leads to oxidative stress which can be responsible for the mitochondrial release of pro-apoptotic factors and cause neuron death. In this regard, it has been reported that during a treatment with caffeine for 7, 14 and 21 days, it completely restored motor function in male Sprague Dawley rats treated with QA. At the same time, the administration of caffeine also significantly reduced oxidative stress in rats treated with QA, thus, increasing endogenous antioxidant capacity and decreasing oxidative damage in a dose-dependent as well as time-dependent manner [123].

On the other hand, it was also reported that high dosages of caffeine, as well as the ablation of adenosine A2A receptor, seem to be detrimental in HD animal models [124].

Finally, one human study conducted in 80 HD male and female patients with an average age of 50 years, showed that caffeine consumption at >190 mg/d over a 10-year period was associated with an earlier age of onset of HD estimated in 1.6 years [120].

Therefore, also for HD, the caffeine effects and adenosine receptor antagonism seem to be highly dose dependent and definitely need further investigation.

## 5. Controlled Release Systems for Caffeine

Caffeine is almost completely absorbed by the gastrointestinal tract within 45 min after oral ingestion, reaching peak plasma concentrations after 0,5 to 2 h. The mean half-life of caffeine in healthy individual plasma has been calculated to be 5 h [10,125].

Since it has a moderately lipophilic structure (experimental logP −0.01, other sources logP −0.07), caffeine can go through the BBB by simple diffusion to exert quick stimulant effects on the CNS [126].

Although caffeine showed interesting results in pre-clinical studies, in clinical trials it has given less impressive outcomes, probably because of the short staying time in plasma and the high speed with which it performs its effects at the level of CNS. Therefore, finding a way to consume caffeine in a form providing the benefits over a longer period may be a solution.

In recent years, there have been produced delivery systems that are able to release caffeine in a controlled and extended way by using conventional or nanotechnological approaches, and, here, we describe the advances of the last 5 years in this field.

A conventional delivery system is the formulation in matrix tablets in which the drug is embedded in an inert material of hydrophilic or hydrophobic polymers. The controlled release is made possible by the presence of polymers that are able to control the diffusion rate of the drug [127]. Kolbina et al. used saturated phosphatidylcholines of stearic and palmitic acids to form the inert matrix hosting caffeine. In this way, there have been produced tablets of low porosity with drug loadings up to 70% that are capable of releasing caffeine by diffusion in an extended and a pH-dependent way [128].

Passing to unconventional approaches, polymeric nanoparticles are largely used for this purpose because polymers have the ability to increase the staying time of nanoparticles in the body without being excreted or detected by the immune system [129]. Caffeine has been entrapped in nano-emulsion droplets stabilized by a polymeric multilayer shell composed of two oppositely charged biopolymers, chitosan and alginate. While chitosan and alginate work synergistically to protect caffeine from oxidation, enzymatic degradation, and hydrolysis, the opposite charges of caffeine and alginate that interact, electrostatically, make easier the entrapping process. Although the encapsulation efficiency is proved to be low (~40%), this delivery system enables the load of high doses of caffeine with a small amount of nano-emulsion [130,131].

In addition to the alginate, chitosan has been complexed to gellan gum to form nanocomplexes encapsulating caffeine by using both bulk and microfluidics methods. Additionally, in this case, the strong electrostatic interactions between chitosan and caffeine allowed drug entrapment with a higher encapsulation efficiency reached with microfluidics methods. The presence of gellan gum provided resistance to low pH and, therefore, the drug release appeared to be more pronounced in intestinal conditions [132].

Caffeine was also encapsulated in different polymeric microparticles by using the freeze-drying technique. Noor et al. used β-glucan, cyclodextrin and resistant starch as biopolymers to compare the encapsulation efficiency and the release rate in simulated gastric and intestinal conditions. The different biopolymers created different microenvironments hosting caffeine. Indeed, in β-glucan microparticles, caffeine is trapped inside a macroporous structure, in those based on β-cyclodextrin, into a hydrophobic cavity and in resistant starch microparticles inside water channels. The study showed that the highest encapsulation efficiency is reached with β-glucan and that the caffeine release, which is more prevalent at neutral pH (intestinal conditions), is more controlled and slower by using β-glucan and resistant starch as carriers [133].

Another delivery system consists of using caffeine loaded nanoparticles based on poly-ε-caprolactone. To make this, the ability of flash nanoprecipitation technique has been tested, which normally is applied to highly hydrophobic molecules (logP > 3.5), to encapsulate a hydrophilic compound such as caffeine. Caffeine was either solubilized in acetone, the solvent, or in water, the anti-solvent, showing that when caffeine is dissolved in water, nanoparticles with improved encapsulation efficiency and loading capacity are produced. In addition, the same nanoparticles showed a better ability to retain caffeine acting as reservoir [134].

The polymeric nanostructures present the right characteristics to cross the BBB and to deliver their encapsulated drug in a controlled and sustained way. However, it appears appropriate to point out their negative implications. Indeed, nanoparticles can give toxicity issues in response to the release of acidic products during their degradation, rendering them inappropriate for extended use in the brain.

Another approach consists of using hydrogels, whose biocompatibility and similar tissue consistency make it suitable for drug delivery. Hydrogels are three-dimensional structures made up of synthetic or natural polymer networks that are capable of absorbing large amounts of water [135].

Milk proteins lend itself to this kind of application since they are biocompatible, biodegradable, non-toxic and have the ability to form hydrogels after forming cross-linking. In their work, Bourbon et al. used lactoferrin and glycomacropeptide as milk proteins to form nanohydrogels encapsulating caffeine. This delivery system showed a high encapsulation efficiency (>90%) and a p-dependent drug release behavior. Indeed, this drug release was ruled by relaxation at pH = 2 and by Fick’s diffusion at pH = 7 [136].

As yeast and fungi cell wall components, chitin-glucan complexes also have suitable characteristics for their use in hydrogel fabrication. Indeed, caffeine has been loaded into chitin–glucan complex hydrogels by using, firstly, a freeze–thaw procedure to facilitate polymer dissolution in NaOH and then a dialysis step to promote gelation. The delivery system behavior appeared different in relation to the kind of solution in which it was. Indeed, the release rate followed Fick’s diffusion when hydrogel was immersed in a PBS solution and a non-Fickian diffusion in a NaOH 0.9% solution [137].

Artusio et al. used, instead, the inverse miniemulsion technique coupled with UV radiation for preparing polymeric nanohydrogels, in which caffeine has been trapped into 100 nm particles, demonstrating how this system provided a controlled and sustained release of the drug within 50 h [138].

In conclusion, since the controlled delivery systems allow to maintain caffeine plasma levels at high for a prolonged time, it would be useful to test their therapeutic potential in neurodegenerative diseases.

## 6. Conclusions and Perspective

The research presented in this review shows that regular coffee consumption, when consumed in moderate quantities, should be able to decrease the overall risk of neurocognitive decline by playing a possible protective action in preventing and delaying the onset of neurodegenerative diseases.

Balanced coffee consumption was linked to slower cognitive decline, mainly, in executive function as determined by neuropsychological tests, as well as a decreased risk of shifting to moderate cognitive impairment. There was no association between coffee consumption and brain atrophy, either in the white matter or in the hippocampus (the area of the brain most involved in AD).

These results would further support the hypothesis that coffee intake may be a highly protective factor, possibly by acting on the neurotoxicity, associated with oxidative stress and mediated by β-amyloid and inflammatory processes. Therefore, there is strong evidence that supports the use of caffeine not as a substitute compound of conventional drug therapies, but rather as a supplement to counteract and contain the neuroinflammatory processes involved in neurodegenerative diseases. Further studies will be indispensable to confirm this result, which once again highlights how the research is also oriented towards eating habits.

## Figures and Tables

**Figure 1 ijms-23-12958-f001:**
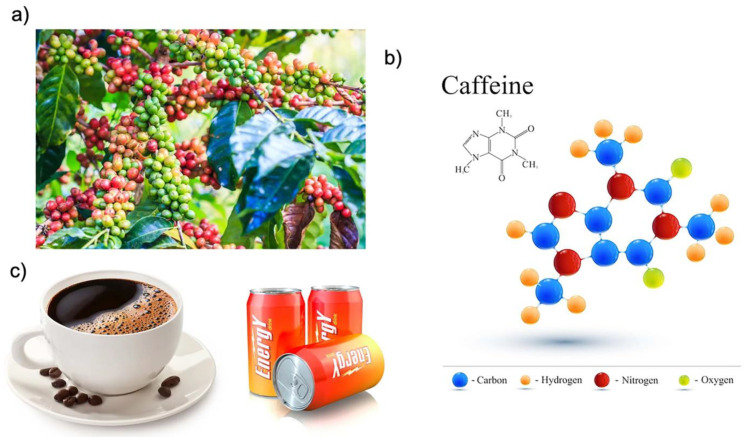
(**a**) Coffee beans ripening on a tree; (**b**) caffeine structure formula; (**c**) drinks containing caffeine.

**Figure 2 ijms-23-12958-f002:**
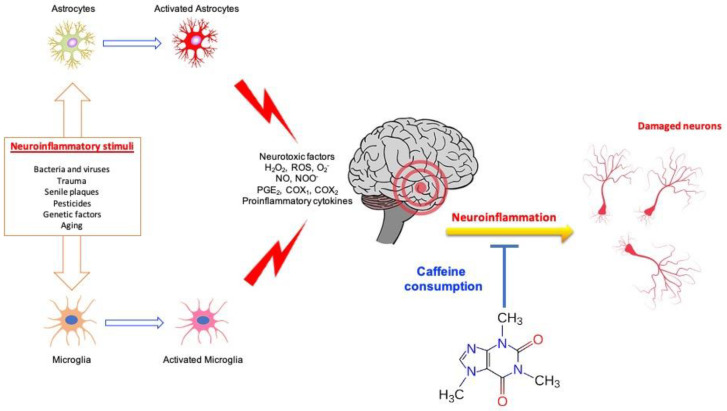
Protective effects of caffeine consumption in a neuroinflammatory condition.

**Figure 3 ijms-23-12958-f003:**
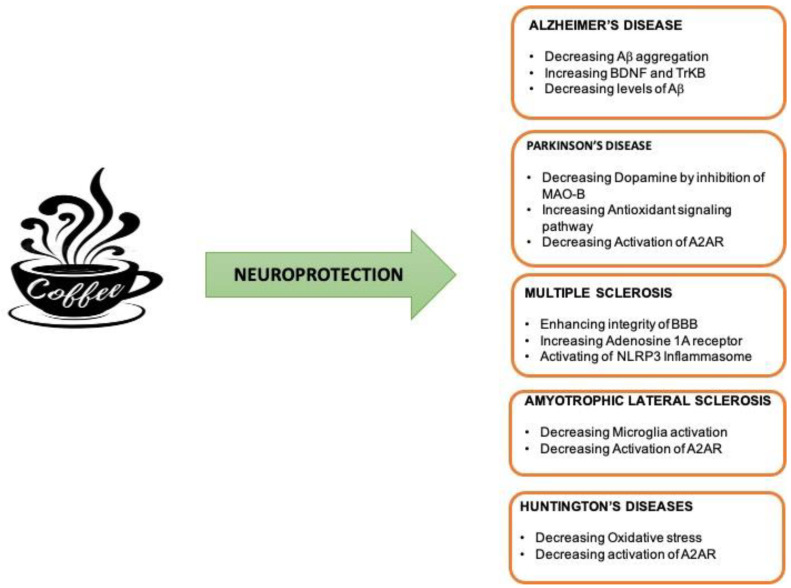
The neuroprotective effects of caffeine in neuroinflammatory and neurodegenerative disease.

## Data Availability

Not applicable.

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
