# Peer review of "Neurodegenerative Diseases: Can Caffeine Be a Powerful Ally to Weaken Neuroinflammation?"

_ijms, 2022, doi:10.3390/ijms232112958_

Round 1

Reviewer 1 Report

In this review the authors summarize the latest research about the potential effects of caffeine in neurodegenerative neurodegenerative disorders prevention, discussing the role of controlled caffeine delivery systems in maintaining high plasma caffeine concentrations for an extended time.
comments
4. Alzheimer's disease also occurs in young people in the genetic variant
4,1The authors report " The evidence on the pathophysiology of this form of dementia is clear”. Actually the pathophysiological mechanism underlying amyloid deposition in brain areas is not known. This concept is also reported in the same chapter by the authors after "the mechanism behind the production of amyloid plaques and neurofibrillary remains unknown."
-The conclusions are too optimistic and certainly coffeine intake alone is not able to improve neurodegenerative diseases. Certainly these in vitro and in vivo animal studies have shown promising results, but no neurologist suggests coffeine therapy.
-The authors should also report possible negative effects of prolonged coffeine intake especially at the level of the cardiovascular system , which may limit the positive effect found in neurodegenerative diseases highlighted in experimental studies.
It is important to explain how the hippocampus, which is the initially involved region, is essential for the regulation of the cortisol/ACTH system, which is certainly involved in the later damage of other brain areas. Cortisol, which usually regulates the stress response in a positive direction, under repeated stress can have an inflammatory effect by damaging other brain areas rich in specific corticosteroid receptors. Coffeine is known to increase the release of cortisol, the main stress-response hormone. Repeated stress is involved in the possible alte damage of the hippocampus. In Figure 2 I would also include cortisol, which is the main stress regulator underlying the mechanisms shown in the figure or I recommend that another figure be included to also explain possible adverse effects of caffeine use in individuals predisposed to neurodegenerative diseases
In conclusion, the review is almost complete by reporting the in vitro or in vivo studies in the animal, but the neurdegenerative diseases whose cause is unknown in almost all cases probably associate genetic and acquired mechanisms that are still mostly unknown

Author Response

We thank the reviewer for his appreciation of the contents of the review and for his comments. In this regard, the aim of our review is not to propose caffeine as a therapeutic substitute for conventional treatments used for the treatment of neurodegenerative diseases.

And it is equally certain that no compound has totally beneficial effects. However, numerous contributions in the literature lean towards positive actions, demonstrated both in vivo and in vitro, attributable to the use of caffeine for the control of inflammatory processes involved in neurodegenerative diseases.

The conclusion of these numerous experimental observations widely documented in the panorama of scientific literature is that its use does not aim to replace standard therapies, but to constitute an additional compound for the containment of the neuroinflammation often associated with these pathologies.

Regarding the fact that, as in many other numerous synthetic and natural compounds, caffeine can have adverse effects, these are cited in the text of the manuscript.

The review talks about moderate levels of caffeine consumption, such as not to be harmful to the body. In this regard, an article has recently been published that highlights this aspect linked to a moderate consumption of caffeine, reporting, in addition, that no correlations were found between poststimulation caffeine or corticosteroid concentrations, and plasticity aftereffects. (10.1016/j.psyneuen.2021.105201).

Reviewer 2 Report

Dear Authors,

the review submitted cover some of the most recent research on the caffeine features against the neuro-inflammation particularly focussing on neurodegenerative disorders. As correctly quoted, the caffeine molecule can be found in many drinks including tea and energy drinks.

Furthermore, the Authors mention the main molecular pathways involving caffeine. 

However, in my opinion some very recent literature should be mention in order to highlight the positive role of caffeine in different neurological disorder (such as the recent review on caffeine and Alzheimer's disease - 10.3390/life12030330) and inflammation together with underlining the negative and detrimental role of beverages (such as energy drinks) on brain.

Indeed, it has been recently demonstrated that caffeine counteract the deleterious effect of LPS-induced retinal inflammation decreasing the cytokine release together with blood retinal barrier preservation (doi: 10.3389/fphar.2021.824885).

On this regard, it has been demonstrated that caffeine does not alter the blood-brain barrier (10.1002/mrm.29355), thus leading to no alteration in the brain parenchyma physiology and homeostasis, maybe due to its lipophilic properties as correctly mentioned in the main text that makes caffeine able to cross the barrier. However, on the other hand, energy drinks are able to induce the blood-brain barrier dysfunction as recently demonstrated (10.3389/fnut.2021.668514), and this side effect could explain the previously observed brain oxidative stress in rats under energy drinks condition (10.1111/1750-3841.13662). 

Thus, the Authors should better distinguish and discriminate between caffeine and different beverages containing caffeine (maybe discussing it in the appropriate section).

Author Response

We thank the reviewer for his comments. Caffeine, as shown in the literature, has multiple effects even if adverse effects have also been described, as often happens with the use of numerous other compounds, both natural and synthetic. The aim of this review is to highlight the anti-inflammatory potential of this compound, as shown by many scientific contributions in the literature, which describe for caffeine a modulating action of neuroinflammation, often associated with neurodegenerative diseases. This beneficial capacity of caffeine does not claim to be a substitute for conventional therapies but goes in the direction of an integrative and supplementary action in the individual's diet in order to improve a possible inflammatory picture associated with neurodegenerative diseases.

Some authors in the literature have also tried to highlight the effects of caffeine as a constituent of beverages containing this compound and it is not our intention to repeat the same comparison (Acta Neurol Scand . 2022 Feb;145(2):127-138), therefore we welcome the suggestion of the review by updating those bibliographic contributions integrating positive and adverse effects of caffeine, as reported in the parts evidenced as changes in the text.
